# Safeguarding in adolescent mental health research: navigating dilemmas and developing procedures

Amie Randhawa [1,2] Grace Wood [1,2] Maria Michail [2,3] Miranda Pallan [4] Paul Patterson,[2,5] Victoria Goodyear [1,2]

¹School of Sport, Exercise and Rehabilitation Sciences, University of Birmingham, Birmingham, UK
²Institute for Mental Health, University of Birmingham, Birmingham, UK
³School of Psychology, University of Birmingham, Birmingham, UK
⁴Institute of Applied Health Research, University of Birmingham, Birmingham, UK
⁵Forward Thinking Birmingham, Birmingham Women's and Children's NHS Foundation Trust, Birmingham, UK

**Correspondence to**
Dr Victoria Goodyear;
v.a.goodyear@bham.ac.uk

## ABSTRACT

Mental disorders are prevalent during adolescence. Self-harm and suicide are more common in adolescents with a probable mental disorder, with one in four reporting to have attempted self-harm. Research involving adolescents is, therefore, likely to include participants experiencing mental ill health, even if mental health is not the primary focus. Researchers should adopt procedures and principles that safeguard adolescent mental health in their research practice. Yet there are gaps between theory and practice of research with adolescents in relation to their mental health, and limited guidance is available.

We discuss emerging safeguarding dilemmas and procedures in adolescent mental health research. Our experiences of safeguarding adolescent mental health are grounded in the UK National Institute for Health and Care Research-funded SMART Schools Study. Drawing from this secondary school-based study, we focus on how our research team encountered and addressed a high prevalence of participants (aged 12–13 and 14–15 years) reporting thoughts and behaviours related to self-harm or suicide (24% of participants). This included reviewing our existing risk mitigation processes and consulting with several committees including young people with lived experiences of mental health.

We present the SMART Schools study safeguarding approach for adolescent mental health. This encompasses key safeguarding principles, study procedures and relevant justifications. We address school and university roles and responsibilities, pupil understanding, and efficient, effective and secure communication pathways. We embed guidance throughout this article for researchers working with adolescents in the context of mental health. Lastly, we present five key recommendations to safeguard the mental health of adolescents participating in research, including (1) appointing a safeguarding lead within the research team; (2) codesigning a bespoke study safeguarding approach; (3) adopting a responsive approach to mental health safeguarding; (4) being transparent about the study mental health safeguarding approach and (5) report the implementation and outcomes of safeguarding approaches.

Trial registration number ISRCTN77948572.

## INTRODUCTION

Mental disorders (eg, anxiety and depression) are prevalent during adolescence (age range 10–19).[1 2] In the UK, it was estimated that one in five adolescents aged 11–16 and one in four adolescents aged 17–19 had a probable mental disorder in 2022.[3] Self-harm and suicide are more common in adolescents with a probable mental disorder: in 2022, 28% of children and adolescents (age 7–16) in the UK with a probable mental disorder reported that they had tried to harm themselves, compared with 2.5% of those who were unlikely to have a mental disorder.[3] Many adolescents with a mental disorder do not receive professional treatment, and this is often related to the stigma associated with mental health, adolescents not disclosing mental health problems and/or a lack of available mental health services and early identification strategies.[4 5] The risks of untreated mental disorders among adolescents are high and include the potential for symptoms to intensify and persist well into adulthood.[5]

Given the high prevalence of adolescent mental disorders, research involving adolescents is likely to include participants experiencing mental ill health, even if mental health is not the primary focus. It is, therefore, expected that researchers should use and be familiar with formal ethical codes and established safeguarding procedures, and have a sufficient level of personal awareness to help them address challenges related to adolescent mental health in their research practice.[6–8] However, there tends to be a gap between the theory and practice of research with adolescents in relation to their mental health.[8 9] A key issue is that although general safeguarding advice to researchers exists,[10–14] there is limited guidance available on specific safeguarding with respect to adolescent mental health, with few documents explicitly mentioning mental health. In turn, researchers are not always prepared to engage with, or appropriately address,

potential adolescent mental health concerns in their research practice.[8 9]

In this article, we discuss emerging safeguarding dilemmas and procedures in adolescent mental health research, drawing on our experiences from a secondary school-based research study. We focus on how our research team encountered and addressed the high proportion of adolescent participants reporting thoughts and behaviours related to self-harm or suicide. To provide support for future research, we present five key recommendations for researchers to safeguard adolescent mental health, including (1) appointing a safeguarding lead within the research team; (2) codesigning a bespoke study safeguarding approach; (3) adopting a responsive approach to mental health safeguarding; (4) being transparent about the study mental health safeguarding approach and (5) report the implementation and outcomes of safeguarding approaches.

## SAFEGUARDING IN ADOLESCENT MENTAL HEALTH RESEARCH: EXPERIENCES FROM THE SMART SCHOOLS STUDY

The experiences of safeguarding adolescent mental health that we discuss in this article are grounded, primarily, in the UK National Institute for Health and Care Research-funded SMART Schools Study (April 2022–July 2025). In this section, we provide a brief overview of the overall study design, further details of which can be accessed from the study protocol.[15]

The aim of the SMART Schools Study is to evaluate the impact of school daytime restrictions on smartphone and social media use on adolescent mental health and well-being. Data are collected from two groups of schools: (1) schools with restrictive smartphone policies that do not permit smartphone use during the school day and (2) schools with permissive smartphone policies that permit smartphone use (eg, during breaks/lunchtimes). A minimum of 1170 pupil participants are planned to be recruited from year 8 (age 12–13 years) and year 10 (age 14–15 years) classes across 30 secondary schools in England (39 adolescents per school).

All procedures of this study were consulted with, and approved by, external groups to the study, including (1) public and patient involvement groups of adolescents, parent/carers and school staff and (2) data monitoring and ethics committee (DMEC) and study steering committee (SSC), consisting of academics and practitioners in the fields of mental health, public health, education, health economics, sports science and epidemiology.

As part of data collection, pupils are asked to complete an online survey which employs validated measures to assess mental well-being (Warwick-Edinburgh Mental Well-being Scale[16]), anxiety (Generalised Anxiety Disorder Assessment[17]) and depressive symptoms (Patient Health Questionnaire-9 (PHQ-9)[18]). Question 9 of the PHQ-9 asks about thoughts of suicide and self-harm and is the central focus of the safeguarding dilemmas that we

encountered within the SMART Schools Study. The question asks participants:

> Over the last two weeks, how often have you been bothered by thoughts that you would be better off dead or of hurting yourself in some way?

Participants are asked to respond on a 4-point Likert scale: (1) not at all, (2) several days, (3) more than half of the days and (4) nearly every day.

## RISK MITIGATION STRATEGIES EMPLOYED TO SAFEGUARD ADOLESCENT MENTAL HEALTH

The SMART Schools Study is an evaluation of a school policy that is already in place, and any risk or harm to participants is not expected as a direct result of a research intervention. Therefore, we elected to obtain active assent from pupils and use an opt-out consent process for pupils' parents/carers. This opt-out parental consent approach respects the autonomy of young people and promotes inclusivity in research (see table 1 for a detailed justification of the consent process). However, there is a possibility that collecting data related to mental health may be a sensitive issue for some participating pupils. Therefore, we developed risk-mitigation strategies to minimise potential risks to the adolescent participants as part of the research, and to ensure effective communication pathways with pupils and parents/carers about the study. An overview of our initial risk mitigation approaches is presented in table 2.

In the context of the PHQ-9 question asking about thoughts of suicide or self-harm, our original risk-mitigation strategy was to monitor pupil responses, and inform schools within 2 weeks if a pupil responded that they had had any thoughts about suicide or self-harm over the previous 2 weeks (Likert scale response options 2–4, ie, several days, more than half of the days or nearly every day). We planned to do this through using notifications within the online survey software (REDCap) that 'flagged' to the research team in real time via email when the concerning response options to question 9 were selected by participants. We planned to communicate this information confidentially and securely to the school so that they could put in place psychological support for these pupils. As communication with the school about individual participant responses would go beyond the limits of confidentiality (which is accepted practice when there is a safeguarding issue), the possibility of disclosure of this information was communicated with pupils prior to them providing assent to participate through the participant information sheets and was reiterated at the beginning of the survey. This information was also communicated with school staff during school recruitment meetings and within the school–university contract, which outlines the expected responsibilities of the school and the research team for the study. Furthermore, the information was communicated to parents through study information sheets as part of the parental opt-out consent process.

**Table 1** SMART Schools Study pupil active assent and parent/carer opt-out consent justification

| Active assent and opt-out consent justification | Explanation |
| --- | --- |
| Adverse effects are not anticipated as a direct result of the study | There is no evidence to suggest that talking about suicide risk and/or experiences could cause harm or induce distress.[21 22] A recent meta-analysis demonstrated that engaging with suicidal research reduces suicidal ideation and behaviours, particularly in high school students.[23] |
| Reducing socioeconomic bias in the sample | Consent letters distributed by schools are less likely to be returned by more socioeconomically disadvantaged parents/carers, therefore, seeking active parental consent through schools is likely to introduce a socioeconomic bias to the sample.[24] |
| Reducing pupil exclusion in the study due to non-engagement by parents rather than active refusal | Research comparing active (opt-in) and passive (opt-out) parental consent obtained through schools for public health-related data collection from pupils aged 11–12 years found a large difference in the proportion of parents providing active and passive consent (41% vs 96%, respectively).[25] This difference between active and passive consent was greater in the most socioeconomically deprived families (31% vs 98% in those in the most disadvantaged quintile, defined by the Index of Multiple Deprivation score). Additionally, the quality of the data obtained from participants was similar for those with active and passive parental consent. A similar observational study[26] also identified that opt-in parental consent was viewed by a headteacher to pose ethical issues, such as excluding the same pupils due to non-engagement by parents. |
| Pupils are of an age of sufficient knowledge and understanding | Pupils are secondary school-age participants (lower age limit=12 years), and at this age, pupils will be able to understand (age appropriate) information given to them about the study and will have the autonomy to decide whether to participate. |

## FURTHER DEVELOPMENT OF A SAFEGUARDING APPROACH FOR SELF-HARM AND SUICIDE

It became apparent during the initial phases of data collection that we needed to further develop our safeguarding procedures for the SMART Schools Study to address the elevated levels of adolescent mental ill health that we were observing in our study sample. In the initial two schools, we collected data from, approximately a quarter of pupils (24%) responded to PHQ-9 question 9 in a way that indicated that they had thoughts of self-harm and/or suicide in the past 2 weeks. In a comparable school-based study with adolescents (aged 13–15) in England, similar findings were observed, with 30% of adolescents indicating that they had thoughts of self-harm.[19] We, therefore, reviewed our existing risk mitigation approaches and agreed that we needed to develop these further to appropriately safeguard adolescents. As part of this process, we consulted with several committees, groups, and individuals to develop safeguarding procedures to support our research in schools.

In a series of meetings, email exchanges and one-to-one follow ups, we engaged with:

► Coinvestigator team for SMART Schools Study: Membership includes; researchers specialising in mental health, self-harm and suicide research, education, public health, and sports science; professionals and practitioners, education and public health researchers, and a qualified teacher and previous headteacher.

► SMART Schools Study Oversight Committees: The DMEC (the role of the DMEC is to monitor the data generated and make recommendations on whether there are any ethical or safety reasons that may influence the study design or conduct and/or the safety, rights and wellbeing of the study participants) and the SSC (the role of the SSC is to provide overall supervision for the study on behalf of the funder and to ensure the study is conducted to the rigorous standards set out in the Department for Health's Research Governance Framework for Health and the Social Care).[20]

► University of Birmingham Governance and Professional Services Departments: Science, Technology, Engineering and Mathematics Research Ethics Committee, information technology security, and legal services.

► School staff: Designated Safeguarding Leads (DSL) at three participating secondary schools.

► University of Birmingham Institute for Mental Health Youth Advisory Group (IMH YAG): Young people (aged 18–25) with lived experiences of mental health.

**Table 2** Initial pupil, school and research team level risk mitigation strategies for safeguarding adolescent mental health

| Pupil | School | Research team |
|---|---|---|
| Pupils to be supervised by teachers when completing the survey | Share pupil and parent/carer participant information sheets in appropriate formats (eg, hard copy letters, email, texts) and inform the research team if materials need translating, to ensure all information reaches pupils and their parents/carers. | Follow university and funder safeguarding and child protection policies and guidance |
| Provide mental health helplines, websites, and guidance for talking to general practitioner in participant information leaflets and at end of pupil survey | Follow the school's safeguarding approach | Complete training courses on mental health and safeguarding children and young people prior to data collection |
| Include a mood-elevation task at the end of the survey[19] | Teachers to support pupils if they become distressed during data collection | Hold an enhanced DBS* |
| The right to withdraw from the study | Use school welfare system for mental health support (where appropriate) | Inform parents of the mental health questions as part of the opt-out consent process |
| | | Inform school staff of any safeguarding issues they become aware of during data collection |
| | | If pupils withdraw during data collection, they are given an alternative task, so their withdrawal is not known to their peers |
| | | Monitor pupil responses to PHQ-9 question 9 and inform schools of any responses other than 'not at all' within 2 weeks of data collection |

*The DBS, which provides a check to ensure researchers are suitable for their role and in relation to working with children.[27]
DBS, Disclosure and Barring Service; PHQ-9, Patient Health Questionnaire-9.

► University of Birmingham Institute for Mental Health Staff: including researchers focused on mental health research.
► Consultant psychiatrist: Expertise in youth mental health.

Informal notes were taken from each of the meetings and emails, and these were consolidated and then discussed by the research team. The feedback from these groups helped us to identify our approach to safeguarding and key safeguarding principles and procedures to address the prevalence of adolescents with self-harm/suicidal thoughts in our sample (see table 3). Overall, the safeguarding approach was based on identifying at-risk adolescents from the pupil survey and then providing that information to schools for specialist or trained teachers (eg, DSL) to then support the identified at-risk adolescents. This approach was grounded in the study design: SMART Schools is a natural experimental observational study (see study protocol for further detail)[15] that involves collecting data related to mental well-being rather than influencing mental health outcomes. An overview of how we apply these key safeguarding principles for the SMART Schools Study is available in online supplemental material 1.

## REVIEW OF THE SMART SCHOOLS SAFEGUARDING APPROACH

To assess the acceptability of the safeguarding approach, we shared the procedure via email with two DSLs from two different schools that were involved in the SMART Schools Study. The DSLs provided feedback on our approach that were integrated into the researcher-focused safeguarding guidelines in the present article. The consensus was that our safeguarding procedures provided timely information related to pupil safeguarding concerns, and facilitated the school's safeguarding system in being responsive to pupils' mental health needs:

DSL 1: I think your approach is sensible and it was helpful from my point of view to hear about concerns on the day so that we could follow up.

DSL 2: I have been really impressed by the safeguarding response from colleagues in the study. The information regarding individual students was provided in-the-moment, which led to better and more meaningful follow-up in terms of our conversations with the children. We were able to record these concerns on our safeguarding system and share with parents who were aware of the study. Thank you for your robust and meaningful safeguarding interventions.

**Table 3** Key safeguarding principles and procedures for adolescent mental health within the SMART Schools Study

| | Key safeguarding principles | SMART Schools Study procedures | Justification for procedures |
|---|---|---|---|
| School and university roles and responsibilities for safeguarding | The research team need a sufficient level of awareness and understanding of safeguarding procedures to be implemented | All research team members to complete safeguarding training via an approved body (the NHS) and the appointment of a study Safeguarding Lead (SL) to oversee safeguarding procedures | The SL will ensure the study safeguarding procedures are implemented |
| | An appropriately trained adult should talk to pupils about their responses to the PHQ-9 questions and/or an adult who has an awareness of pupil pre-existing mental health problems and/or contextual factors that may be influencing a pupil's mental health (eg, family/home circumstances, exams) | Research team to access the name and contact details of the School Designated Safeguarding Lead (DSL) to ensure they are aware of the study and to confirm the school will provide support for pupil mental health, where necessary | All schools will have a DSL, as this is a statutory role. The school DSL is the staff member with overarching responsibility and training for pupil mental health, as well as awareness of appropriate and trusted school staff to support pupil mental health needs in the school. They will also have access to information on individual pupil mental health needs |
| | Schools should have sufficient information on how to support pupil mental health | Research team to provide schools with a mental health resource pack, which includes support for directing pupils to mental health resources and GP referral services | The resource pack will ensure that all schools are provided with evidence-based resources |
| Pupil understanding | Ensure pupils have a sufficient level of understanding about the PHQ-9 Question 9, and that they know that their responses may be shared with teachers in the school | An overview of PHQ-9 questions, and that information may be communicated to teachers, is shared with pupils in written (pre-distributed participant information leaflets and at the start of the survey) and verbal (in class before data collection) formats | Communicating the information in multiple formats at several time points maximises the chances of information reaching pupils and provides opportunities and time for pupils to ask questions |
| Efficient, effective and secure communication pathways | Information about pupil responses to PHQ-9 question 9 must reach schools in sufficient time to enable teachers to respond on the same day | DSL emailed with dates, times and classes for data collection and provided with details of PHQ-9, 7 days in advance | Communication of information about the PHQ-9 Question 9 prior to, during and immediately following data collection aims to support schools in being able to respond to pupils' mental health needs in a time efficient manner |
| | Pupil responses to questions must be communicated with the school in a way that is legally compliant, such as following General Data Protection Regulations | DSL (or relevant member of staff) is verbally notified in the school of pupil PHQ-9 question 9 responses | Verbal communication and receipting of emails ensures that schools have timely access to appropriate information to support their pupils' mental health |
| | | DSL is emailed on the day of data collection with the names of pupils who raised concerns with responses to PHQ-9 Question 9, in an encrypted, secure file transfer | Using secure file transfer and encryption for sensitive personal data complies with legal frameworks and school and university data security requirements |
| | | DSL required to receipt email within 24 hours, and if not receipted the research team phones school for confirmation | |
| | | Data collection to take place Monday to Thursday | If data collection was completed on Fridays, schools may not have sufficient time to respond to pupils' mental health needs before the weekend and where negative thoughts could intensify |

GP, general practitioner; NHS, National Health Service; PHQ-9, Patient Health Questionnaire-9.

We also consulted with the University of Birmingham Institute for Mental Health's Youth Advisory Group. Their feedback was:

> Using our recent experiences of being school pupils, we provided feedback on the safeguarding approach that had been put before us and put forward suggestions on how the approach could be made more effective. These changes have been integrated into the safeguarding approach, which we feel is acceptable and will be effective because it:
>
> • Monitors and supports mental health issues
>
> • Involves trusted adults
>
> • Provides a safeguarding process that is clear to young people
>
> (IMH YAG Members)

## RECOMMENDATIONS FOR SAFEGUARDING IN ADOLESCENT MENTAL HEALTH RESEARCH

Based on the SMART Schools Study safeguarding experiences, we have embedded guidance throughout this article for researchers working with adolescents in the context of mental health. In summary, we suggest five key recommendations for researchers and future research planning in relation to safeguarding in adolescent mental health and well-being research:

1. Appoint a safeguarding lead to the research team: Identify a member of the research team who is responsible for designing, reviewing, implementing and communicating the study safeguarding procedures.
2. Codesign a bespoke study safeguarding approach: Develop and implement a safeguarding approach specific to adolescent mental health. This includes codesigning the safeguarding approach with adolescents who have lived experiences of mental ill health (eg, Youth Advisory Groups), representatives from schools (eg, pupils and school staff) and members of the research team (eg, SMG), while using evidence from existing recommendations, best practices or research findings in a similar context. This use of codesign and prior literature would ensure that the contextual needs of participants and schools are considered and integrated within the research in an evidence-based way. In some studies, particularly, those with a mental health focus, it may be appropriate to codevelop a wellness plan that would include details of a support person and a safety plan, and this would be completed with young people prior to study entry.
3. Adopt a responsive approach to safeguarding: Be responsive and adaptable to unexpected events, ensuring that appropriate adjustments are made that reflect the needs of the participants and contexts. This includes actively seeking out appropriate expertise to develop, revise and implement contextually relevant safeguarding procedures.
4. Be transparent about your safeguarding approach: Ensure that the study safeguarding procedures, and how they are implemented, are transparent and understood by all involved in or contributing to the study. This includes identifying and establishing effective communication pathways between pupils, parents/carers, school staff and researchers at the onset of the research.
5. Report the implementation and outcomes of safeguarding approaches: Researchers should report the implementation and outcomes of different safeguarding approaches so that the research community can learn from these, thus improving safeguarding procedures further over time.

**Acknowledgements** We would like to thank the following for their involvement and input in the development of the safeguarding procedures: SMART Schools Study team; Study Management Group members Professor Peymane Adab, Professor Hareth Al-Janabi, Dr Alice Sitch, Dr Sally Fenton, Dr Paul Patterson, Dr Matthew Wade, Dr Breanna Morrison, Ms Kirsty Jones (Study Management Group); Study Steering Committee members Professor Lorraine Cale, Professor Scott Leatherdale, Dr Barbara Barrett, Dr Samuel Relton, Mr Steve Cotton; and Data Monitoring and Ethics Committee members Dr Thomas Quarmby, Professor Chris Owen and Dr Tessa Reardon. We would also like to thank the Institute for Mental Health Youth Advisory Group and Designated Safeguarding Leads who engaged with reviewing and providing feedback on our safeguarding approach.

**Contributors** VG and MP are chief investigators and obtained funding for the study. AR, GW and VG drafted the manuscripts, with critical input from MM, MP and PP. All authors reviewed and approved the final draft.

**Funding** This project is funded by the NIHR Public Health Research programme (NIHR131396).

**Disclaimer** The views expressed are those of the author(s) and not necessarily those of the NIHR or the Department of Health and Social Care.

**Competing interests** None declared.

**Ethics approval** This study involves human participants and full ethical approval was obtained from the University of Birmingham's Science Technology, Engineering and Mathematics Research Ethics Committee on 8 July 2022 (ERN_22-0723).

**Provenance and peer review** Not commissioned; externally peer reviewed.

**ORCID iDs**
Amie Randhawa http://orcid.org/0000-0002-3770-2028
Grace Wood http://orcid.org/0000-0003-2622-2501
Maria Michail http://orcid.org/0000-0001-7380-3494
Miranda Pallan http://orcid.org/0000-0002-2868-4892

Victoria Goodyear http://orcid.org/0000-0001-5045-8157

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
