## [Reviewer comments · BMJ Open]

ARTICLE DETAILS

TITLE (PROVISIONAL)	Safeguarding in Adolescent Mental Health Research: Navigating Dilemmas and Developing Procedures
AUTHORS	Randhawa, Amie; Wood, Grace; Michail, Maria; Pallan, Miranda; Patterson, Paul; Goodyear, Victoria

VERSION 1 – REVIEW

REVIEWER	Dow, Dorothy Duke University Medical Center, Pediatrics
REVIEW RETURNED	10-Sep-2023

GENERAL COMMENTS	Safeguarding in Adolescent Mental Health Research: Navigating Dilemmas and Developing Procedures Wood, G. E. R.1,2, Randhawa, A. E.1,2, Michail, M. 2,3, Pallan, M.4, Patterson, P.2,5, Goodyear, V.A.* 1,2 Peer Review Report The authors describe a need for guidelines to safeguard adolescent mental health due to the high prevalence of mental health conditions in adolescents, and therefore the high likelihood that adolescents recruited in research studies have mental health illnesses. This study jumps on an existing study titled The SMART Schools Study on the effect of restrictive versus permissive smartphone policies on adolescent mental health in England, to improve safeguarding precautions after initial data collection highlighted the high prevalence of at risk adolescent participants. The study does bring up an important point where there is a higher prevalence of mental health disorders in adolescents, and there is a need for regulated guidelines to address and help youth whose safety concerns are discovered during a research study. The recommendations that have been suggested are relevant to establishing a guideline for safeguarding adolescents' mental health in research. However, there are a few major concerns regarding this study: 1. While the introduction highlights a potential gap in the literature, there is a need to further elaborate on the evidence that previous researchers working with adolescents have also faced difficulties when not having safeguarding principles in place, and that there is a call for these safeguarding principles.2. There is an inadequate explanation of the methodology of identifying the key safeguarding principles (Page 7 of 15, line 39 onwards). For instance, how many people were involved in providing feedback for the safeguarding principles? How was feedback obtained from the various parties? Was it via meetings, or email exchanges? Providing examples of the qualitative data to substantiate the suggested guidelines will be helpful. Was there thematic analysis done on the feedback given to generate the
---

	points? Is there any evidence in the existing literature that these suggested guidelines will prove helpful to the adolescents? 3. Table 3, which has a typo in the title, is an important table that is supposed to outline the safeguarding principles and planned procedures that should be implemented, however certain points can be elaborated on. For instance, what does the safeguarding training include? Has this been done before in the literature? What is included in the mental health resource pack? How are the DSLs elected? Included in the supplementary material 1 is how these safeguarding principles will be applied. Again, it can be improved with more details. Such as, what if the school does not already have a DSL, when would the process of electing one start, and what is included in the SMART Schools Study Mental Health resource letter? This can be included in a supplementary material. Another key question is what happens once the DSL acknowledges the safety concern? This is important as this is ultimately what would help youths with mental illnesses discovered in research studies. If the plan is to follow the school's approach to addressing mental health concerns, is there a need to oversee or obtain some accountability for this as well? 4. In terms of the review of the SMART schools safeguarding approach (Page 10 of 15), again a more vigorous analysis of comments from the various parties will be helpful. Detailed methods of obtaining data will be good as well, with how many DSLs provided comments, how feedback was obtained. Coding of themes from the data will be helpful. Only positive comments were included in the examples given, and I am wondering if there are comments highlighting improvements to address. I am also curious to know why only the DSLs and YAG gave feedback regarding these guidelines, and is it possible to collect comments from other parties involved in the project, namely the youths, or caregivers? This would prove for a more robust support that these suggested guidelines are helpful. 5. The conclusion mentions the four recommendations for future research with adolescents. And while the suggestions are relevant to addressing the issue, it would be beneficial to include evidence from existing literature, if some of these methods have been included before. Furthermore, the authors have omitted limitations of the study as well as areas for future exploration. Limitations of this study can include the lack of evidence from the adolescents or caregiver in this study that this suggested safeguarding approach works, and the need to explore the effects of implementing these approaches in a controlled study with more objective outcomes. Furthermore, the study ends at the implementation phase of these guidelines, and does not elaborate on what happens after a DSL has acknowledged receipt of the safety concern. It will be helpful to include what is currently being done at the schools once the responsibility has been given to the DSL, and highlight if this is helpful to the adolescents, and whether or not this management downstream needs to be regulated. Some minor changes to consider as well:  1. There are two Table 1's. Please change the Safeguarding principles to Table 3. Please also spell out abbreviations used in the table in a footnote under the table. 2. The title of Table 2 can be phrased clearer to let readers know that this was the initial risk mitigating strategies for the SMART study. 3. It is unclear if it is 2 weeks or same day that an adolescent endorsing mental health symptoms is expected to discuss
--	---

	symptoms (line 25 page 5) or same day (key safeguarding principles in table “3” In summary, while this study is relevant to address a need in studies related to adolescent mental health, it can be improved with a more detailed methodologies section and vigorous qualitative analysis, with more elaboration on current dilemmas faced by researchers and future recommendations. This review was completed with support from 3rd year Medical Student supporting adolescent mental health research, Dana Chow Wai Shin
--	---

REVIEWER	McKay, Sam Orygen The National Centre of Excellence in Youth Mental Health
REVIEW RETURNED	01-Nov-2023

GENERAL COMMENTS	I have reviewed the manuscript titled “Safeguarding in Adolescent Mental Health Research: Navigating Dilemmas and Developing Procedures. It provides a helpful overview of potential directions for improved safeguarding in mental health research with young people based on approaches adopted in the SMART study. There are two issues that I think would be beneficial to address before publication, which I have outlined below. The second recommendation for safeguarding in adolescent mental health research suggests that a co-designed bespoke approach should be utilized in each study. While I agree that context-specific approaches are optimal, it could be beneficial to highlight the possibility of basing this process on existing recommendations, best practice, or research findings from similar contexts. This would minimise the need to reinvent similar bespoke approaches for each study and allow for safeguarding approaches to be further developed and refined based on ongoing recommendations and research. Building on the above point, I wonder whether it may be worth including a fifth recommendation of reporting on the implementation and outcomes of different safeguarding approaches so that the research community can use learning within the community to improve safeguarding further over time.
---

VERSION 1 – AUTHOR RESPONSE

Reviewer Comment	Response
N/A	Please note, we have changed the author order for this article because one of the authors has now left the study team and no longer wishes to be lead author.
Reviewer 1 (in text changes in red text)	
The authors describe a need for guidelines to safeguard adolescent mental health due to the high prevalence of mental health conditions in	We would like to thank you for your comments regarding our article on adolescent

adolescents, and therefore the high likelihood that adolescents recruited in research studies have mental health illnesses. This study jumps on an existing study titled The SMART Schools Study on the effect of restrictive versus permissive smartphone policies on adolescent mental health in England, to improve safeguarding precautions after initial data collection highlighted the high prevalence of at risk adolescent participants. The study does bring up an important point where there is a higher prevalence of mental health disorders in adolescents, and there is a need for regulated guidelines to address and help youth whose safety concerns are discovered during a research study. The recommendations that have been suggested are relevant to establishing a guideline for safeguarding adolescents' mental health in research. However, there are a few major concerns regarding this study:	safeguarding. In this table we outline how we have addressed each of the comments. We feel that it is important to initially clarify that this article is a 'Communication Article', the guidance for which on the BMJ website states: "Rather than presenting primary research, it is an opportunity to present ideas, examples, and innovations relating to the conduct of clinical research." (see Introducing BMJ Open's new article type: The Communication Article - BMJ Open). As such, the submitted article does not report on outcomes from a study, and instead discusses how, with the involvement of patients and the public (PPI), and in response to the experience of researching in schools, we have co-developed a safeguarding approach, which we put into practice in our study carried out with adolescents in school settings in England (SMART Schools Study). We feel that this is important to emphasise as our article submission category has relevance to many of the subsequent comments and responses.
While the introduction highlights a potential gap in the literature, there is a need to further elaborate on the evidence that previous researchers working with adolescents have also faced difficulties when not having safeguarding principles in place, and that there is a call for these safeguarding principles.	The issue in focus (high prevalence of safeguarding concerns) emerged in the conduct of our research, and the research team sought out safeguarding guidance on how to address this, but we were unable to uncover any guidance that was immediately relevant to this issue (as discussed in paragraph 2, page 2). However, we do highlight in the article that "...researchers are not always prepared to engage with, or appropriately address, potential adolescent mental health concerns in their research practice (8, 9)." (p2).
There is an inadequate explanation of the methodology of identifying the key safeguarding principles (Page 7 of 15, line 39 onwards). For instance, how many people were involved in providing feedback for the safeguarding principles? How was feedback obtained from the various parties? Was it via meetings, or email exchanges? Providing examples of the	As this is a responsive communications article, grounded in experience and in the PPI activity involved in co-producing our safeguarding approach, the methodology focuses on key learning points that were gathered from the experience of our schools researchers and then refined through iterative discussions and feedback with stakeholders. Within the article

qualitative data to substantiate the suggested guidelines will be helpful. Was there thematic analysis done on the feedback given to generate the points? Is there any evidence in the existing literature that these suggested guidelines will prove helpful to the adolescents?	we have provided detail on who was consulted, and their personal or professional backgrounds. We have added detail on how the feedback was obtained, on p7: In a series of meetings, email exchanges, and one to one follow ups, we engaged with: We have also added in the following to clarify how we managed the feedback (p7): Informal notes were taken from each of the meetings and emails, and these were consolidated and then discussed by the research team.
Table 3, which has a typo in the title, is an important table that is supposed to outline the safeguarding principles and planned procedures that should be implemented, however certain points can be elaborated on. For instance, what does the safeguarding training include? Has this been done before in the literature? What is included in the mental health resource pack? How are the DSLs elected?	We have elaborated on several points within this table (on page 8). We have added in details about the safeguarding training in both Table 2 and Table 3. In Table 2 we have added that the safeguarding training is about safeguarding children and young people, and in Table 3 we have added that it was run by an approved body (the NHS), therefore indicating that this was a professionally run course, available to other researchers. We have added in further detail about the Mental health resource pack: which includes support for directing pupils to mental health resources and GP referral services It is outside the remit of this communication article to elaborate on the selection process of DSLs. This is a fundamental school role and as such it is a requirement that all schools have a fully trained DSL. Often the role is taken by a member of the individual school's senior leadership team such as a Headteacher or Deputy Headteacher.

Included in the supplementary material 1 is how these safeguarding principles will be applied. Again, it can be improved with more details. Such as, what if the school does not already have a DSL, when would the process of electing one start, and what is included in the SMART Schools Study Mental Health resource letter? This can be included in a supplementary material.

Another key question is what happens once the DSL acknowledges the safety concern? This is important as this is ultimately what would help youths with mental illnesses discovered in research studies. If the plan is to follow the school's approach to addressing mental health concerns, is there a need to oversee or obtain some accountability for this as well?

With regard to Supplementary Material 1 all schools already have to appoint a DSL as a statutory role. The SMART Schools Study Mental Health resource letter referred to is the mental health resource pack discussed above. The wording in this table has been modified so that it is clear that this is the same document (and what it includes).

Regarding what happens once a school DSL acknowledges the safety concern, and our accountability, we have added the following text to page 7:

Overall, the safeguarding approach was based on identifying at-risk adolescents from the pupil survey and then providing that information to schools for specialist or trained teachers (e.g., Designated Safeguarding Leads [DSL]) to then support the identified at-risk adolescents. This approach was grounded in the study design: SMART Schools is a natural experimental observational study (see Wood et al. 2023 for further detail) that involves collecting data related to mental wellbeing rather than influencing mental health outcomes.

In addition, Table 3 outlines that an appropriately trained adult should talk to the pupils about their responses, and one who has some prior knowledge about pre-existing problems and contextual factors. The table highlights that the school DSL would be best placed to support this process as this is a statutory role for schools and each individual school follows its own internal safeguarding policy protocols that are in turn based on national policy and guidance from the UK Government Department of Education. The critical issue addressed by our paper is the dearth of safeguarding guidance focusing on the experience of researchers working in school settings as any such guidance must be both flexible enough to safely interface with the broad range of individual school safeguarding policies

	whilst being helpful and supportive to research professionals.
In terms of the review of the SMART schools safeguarding approach (Page 10 of 15), again a more vigorous analysis of comments from the various parties will be helpful. Detailed methods of obtaining data will be good as well, with how many DSLs provided comments, how feedback was obtained. Coding of themes from the data will be helpful. Only positive comments were included in the examples given, and I am wondering if there are comments highlighting improvements to address. I am also curious to know why only the DSLs and YAG gave feedback regarding these guidelines, and is it possible to collect comments from other parties involved in the project, namely the youths, or caregivers? This would prove for a more robust support that these suggested guidelines are helpful.	We have included detail on the DSLs approached, and the methods of obtaining the data (p9): To assess the acceptability of the safeguarding approach, via email with two DSLs from two different schools that were involved in the SMART Schools study. The DSL's provided feedback on our approach that were integrated into the researcher focused safeguarding guidelines in the present article. As the SMART Schools project is not an intervention-based study, our field researchers are not clinically trained. As such we agreed that it was appropriate to work with individuals who are trained and/or are experts in young peoples' mental health and wellbeing. For the purposes of the current safeguarding article, we therefore sought feedback from these parties.
The conclusion mentions the four recommendations for future research with adolescents. And while the suggestions are relevant to addressing the issue, it would be beneficial to include evidence from existing literature, if some of these methods have been included before. Furthermore, the authors have omitted limitations of the study as well as areas for future exploration. Limitations of this study can include the lack of evidence from the adolescents or caregiver in this study that this suggested safeguarding approach works, and the need to explore the effects of implementing these approaches in a controlled study with more objective outcomes. Furthermore, the study ends at the implementation phase of these guidelines, and does not elaborate on what happens after a DSL has acknowledged receipt of the safety concern. It will be helpful to include what is currently being done at the	We agree and have elaborated on the recommendations and in addition incorporated the relevant comments on this element from reviewer 2 (see below). As indicated in the introduction to the piece, there was found to be a gap in the literature, and so our submitted communications article emerged from this context. We have not included limitations because this is a communications piece, grounded in public involvement in research (PPI), and not a report of a specific research study into the development of adolescent safeguarding processes in research. The aim is to provide guidance, co-designed by researchers and key

schools once the responsibility has been given to the DSL, and highlight if this is helpful to the adolescents, and whether or not this management downstream needs to be regulated.	stakeholders, to benefit future researchers in the area.
Some minor changes to consider as well:  1. There are two Table 1's. Please change the Safeguarding principles to Table 3. Please also spell out abbreviations used in the table is a footnote under the table. 2. The title of Table 2 can be phrased clearer to let readers know that this was the initial risk mitigating strategies for the SMART study. 3. It is unclear if it is 2 weeks or same day that an adolescent endorsing mental health symptoms is expected to discuss symptoms (line 25 page 5) or same day (key safeguarding principles in table "3") In summary, while this study is relevant to address a need in studies related to adolescent mental health, it can be improved with a more detailed methodologies section and vigorous qualitative analysis, with more elaboration on current dilemmas faced by researchers and future recommendations. This review was completed with support from 3rd year Medical Student supporting adolescent mental health research, Dana Chow Wai Shin	Thank you for highlighting this error, however, there appears to be a table numbering issue when the file is uploaded. On the version we have uploaded it is labelled 'Table 3' – but following the upload it somehow changes to 'Table 1'. We cannot find a way around this issue. Abbreviations have been added to the bottom of the table (p8). The title of this table has been edited to include the word 'initial' (p6) and this has also been included in the description of the table (p4). Thank you for highlighting this. Table 3 has now been edited to include the wording that the school DSL will be notified on the same day as data collection. In the initial approach it was within 2 weeks, which should be clearer now the title has been amended as in the point above. As described above this submission was not planned as a research article but rather a communications piece, grounded in the PPI activity involved in co-producing our safeguarding approach. Further elaboration was provided where necessary, but more detailed methodology and analysis sections were not deemed appropriate given the Journal's communications criteria.
Reviewer 2 (in text changes in blue text)	

Comments to the Author: I have reviewed the manuscript titled “Safeguarding in Adolescent Mental Health Research: Navigating Dilemmas and Developing Procedures. It provides a helpful overview of potential directions for improved safeguarding in mental health research with young people based on approaches adopted in the SMART study. There are two issues that I think would be beneficial to address before publication, which I have outlined below.	Thank you for your comments. In this table we outline how we have addressed each of the comments.
The second recommendation for safeguarding in adolescent mental health research suggests that a co-designed bespoke approach should be utilized in each study. While I agree that context-specific approaches are optimal, it could be beneficial to highlight the possibility of basing this process on existing recommendations, best practice, or research findings from similar contexts. This would minimise the need to reinvent similar bespoke approaches for each study and allow for safeguarding approaches to be further developed and refined based on ongoing recommendations and research.	This has been added into the second recommendation on p10: This includes co-designing the safeguarding approach with adolescents who have lived experiences of mental ill-health (e.g. Youth Advisory Groups), representatives from schools (e.g. pupils and school staff), and members of the research team (e.g. SMG), while utilizing evidence from existing recommendations, best practices, or research findings in a similar context. This use of co-design and prior literature would ensure that the contextual needs of participants and schools are considered and integrated within the research in an evidence-based way.
Building on the above point, I wonder whether it may be worth including a fifth recommendation of reporting on the implementation and outcomes of different safeguarding approaches so that the research community can use learning within the community to improve safeguarding further over time.	We thank you for this helpful suggestion and have added it as an additional recommendation on p10: 5. Report the implementation and outcomes of safeguarding approaches: Researchers should report the implementation and outcomes of different safeguarding approaches so that the research community can learn from these, thus improving safeguarding procedures further over time. To reflect this, slight edits have also been made in the Abstract (p1) and on pages 2 and 9.

VERSION 2 – REVIEW

REVIEWER	McKay, Sam Orygen The National Centre of Excellence in Youth Mental Health
REVIEW RETURNED	06-Dec-2023
GENERAL COMMENTS	I have reviewed the revised manuscript. The authors have effectively addressed my prior minor concerns and from my perspective the article manuscript is acceptable for publication.